# Efficacy of Nanofiber Sheets Incorporating Lenvatinib in a Hepatocellular Carcinoma Xenograft Model

**DOI:** 10.3390/nano12081364

**Published:** 2022-04-15

**Authors:** Terufumi Yoshida, Masaki Kaibori, Nanami Fujisawa, Mariko Ishizuka, Fusao Sumiyama, Masahiko Hatta, Hisashi Kosaka, Kosuke Matsui, Kensuke Suzuki, Tomoya O. Akama, Tayo Katano, Kengo Yoshii, Mitsuhiro Ebara, Mitsugu Sekimoto

**Affiliations:** 1Department of Surgery, Kansai Medical University, 2-5-1 Shinmachi, Hirakata 573-1010, Japan; yoshiter@hirakata.kmu.ac.jp (T.Y.); ishizukm@takii.kmu.ac.jp (M.I.); sumiyamf@hirakata.kmu.ac.jp (F.S.); hattamas@hirakata.kmu.ac.jp (M.H.); kosakahi@hirakata.kmu.ac.jp (H.K.); matsuik@hirakata.kmu.ac.jp (K.M.); sekimotm@hirakata.kmu.ac.jp (M.S.); 2Research Center for Functional Materials, National Institute for Materials Science (NIMS), Tsukuba 305-0044, Japan; nfujisawa.tokyo@gmail.com (N.F.); ebara.mitsuhiro@nims.go.jp (M.E.); 3Department of Otolaryngology, Head and Neck Surgery, Kansai Medical University, Hirakata 573-1010, Japan; suzukken@hirakata.kmu.ac.jp; 4Department of Pharmacology, Kansai Medical University, Hirakata 573-1010, Japan; akamat@hirakata.kmu.ac.jp; 5Department of Medical Chemistry, Kansai Medical University, Hirakata 573-1010, Japan; katanot@hirakata.kmu.ac.jp; 6Department of Mathematics and Statistics in Medical Sciences, Kyoto Prefectural University of Medicine, Kyoto 606-0823, Japan; yoshii-k@koto.kpu-m.ac.jp

**Keywords:** lenvatinib, poly ε-caprolactone, electrospun nanofiber sheet, HuH-7, drug delivery system

## Abstract

Lenvatinib has a high response rate in unresectable advanced hepatocellular carcinoma (HCC). In this study, we investigated whether lenvatinib-incorporating poly(ε-caprolactone) sheets (lenvatinib sheets) as a drug delivery system (DDS) exerted antitumor effects in a murine HCC model. The lenvatinib sheets were designed for sustained release of approximately 1 mg lenvatinib for 14 days. For 14 days, 1 mg lenvatinib was orally administered to mice. Then, we compared the antitumor effects of lenvatinib sheets with those of oral lenvatinib. The tumor volume, body weight, and serum lenvatinib level were measured for 14 days. A peritoneal dissemination model was established to examine the survival prolongation effect of the lenvatinib sheets. Tumor growth was significantly inhibited in the lenvatinib sheet group compared with that in the no treatment and oral groups. The antitumor effect was significantly higher in the lenvatinib sheet group. Regardless of the insertion site, the serum lenvatinib levels were maintained and showed similar antitumor effects. The mitotic index was significantly inhibited in the lenvatinib sheet group compared with that in the control group. Furthermore, lenvatinib sheets improved the 30-day survival. Lenvatinib sheets showed sufficient antitumor effects and may serve as an effective novel DDS for advanced HCC.

## 1. Introduction

Hepatocellular carcinoma (HCC) is the sixth most common cancer and was the third most common cause of cancer-related death worldwide in 2020 [1]. Although most affected patients are in East Asia and Africa, the incidence and mortality rates of HCC are increasing in various parts of Europe and the USA [2]. In Japan, most cases of HCC are detected in patients with chronic hepatitis and liver cirrhosis induced by infection with hepatitis B or C virus. The first molecular-targeted drug for unresectable HCC (u-HCC), sorafenib, was developed in 2009 as a first-line therapy based on the results of the SHARP [3] and Asia-Pacific [4] trials. Following the development of sorafenib, lenvatinib was approved as another first-line agent in 2018, following the results of the REFLECT trial [5]. Furthermore, in September 2020, atezolizumab plus bevacizumab therapy, a combination of an immune checkpoint inhibitor and anti-vascular endothelial growth factor (VEGF), was introduced as a first-line treatment option for u-HCC [6]. Lenvatinib is an oral multi-kinase inhibitor that selectively inhibits signal transduction mainly through VEGF receptors 1–3, fibroblast growth factor receptors 1–4, platelet-derived growth factor receptor α, rearranged during transfection receptor tyrosine kinase, and KIT [7]. It has a high response rate (37.0–40.6%) as a major anticancer drug for advanced u-HCC. However, its systemic adverse effects (AEs), such as hypertension, proteinuria, and weight loss, are prominent, forcing 39.9–47.8% of patients to interrupt treatment, 36.9–73.9% to reduce the dose, and 8.8–21.7% to discontinue the drug. Grade ≥3 AEs occurred in 54–57% of affected patients, and the risk of AEs could affect the treatment outcome [5,7,8]. Although the survival rate of patients with HCC is improving owing to advances in early diagnosis and multidisciplinary treatment, many patients with HCC have a background of liver disorders, such as cirrhosis and chronic hepatitis, or are refractory to treatment and are often unable to continue effective anticancer drug therapy, having to reduce or discontinue treatment.

In cancer chemotherapy, there are two types of drug delivery systems (DDS): targeting DDS, such as liposome- [9] and micelle-based [10] systems and antibody-drug conjugates (ADCs) [11], developed to deliver drugs only to the local tumor area using their enhanced permeability and retention [12], and controlled release DDS, such as sustained-release microcapsule drug systems [13] such as Leuplin^®)^. Currently, no DDS formulations have been developed for the long-term sustained release of drugs with high antitumor efficacy.

Poly(ε-caprolactone) (PCL) is a biodegradable polymer approved by the Food and Drug Administration for use in various biomaterial applications [14,15]. PCL and drugs can be easily converted into nanofibers by electrospinning using common solvents, and the length, thickness, and strength of the drug-encapsulated fibers can be customized [14,15,16]. The drug incorporated in a sheet composed of these fibers can be released locally and quantitatively by diffusion [15,17,18].

In this study, we investigated whether a PCL nanofiber sheet incorporating lenvatinib (lenvatinib sheet) as a DDS can effectively and continuously deliver a high systemic antitumor effect of lenvatinib to liver cancer in a murine xenograft model.

## 2. Materials and Methods

### 2.1. Fabrication of Electrospun PCL Nanofiber Sheets

PCL, a biocompatible polymer, was used to prepare fibers for sustained release of lenvatinib. As previously described [19], PCL (molecular weight; 80,000) and lenvatinib were co-dissolved in 1,1,1,3,3,3-hexafluoro-2-propanol. The solution ratios were 20 w/v% for PCL and 5 w/v% for lenvatinib. The solution was placed in a syringe with a 24-gauge needle and then subjected to 20 kV and ejected across a 13-cm syringe-collector gap at a flow rate of 1.0 mL/h (Nanon-01A, MECC Co., Ltd., Fukuoka, Japan). Fiber morphology was characterized using a scanning electron microscope (JCM-5000, JEOL, Tokyo, Japan). The diameters of the nanofibers were measured using ImageJ software (Figure 1A). The thermal properties of the PCL nanofibers were measured by differential scanning calorimetry (DSC) (DSC600, HITACHI, Tokyo, Japan) at a 15 °C min^−1^ programming rate by using alumina pans (Appendix A). 

### 2.2. Drug Release Profile

To perform the drug release study, sheets containing 20 mg of lenvatinib (*n* = 3) were immersed in 5 mL of phosphate-buffered saline (PBS) and placed in an incubator at 37 °C with stirring at 100 rpm to release lenvatinib. PBS containing the released drug was collected every 24 h, and the process of immersing the fibers in 5 mL of fresh PBS was repeated for 56 days. The amount of lenvatinib released in the collected PBS was calculated by measuring the absorption intensity at 250 nm, which is the absorption wavelength specific to lenvatinib, using a NanoDrop^TM^ 2000 spectrophotometer (Thermo-Fisher Scientific, Waltham, MA, USA).

### 2.3. Cell Lines and Reagents

The human hepatocyte cell lines HuH-7 and HuH-7-Luc were obtained from the Japanese Collection of Research Bioresources Cell Bank (Osaka, Japan). The cells were incubated at 37 °C in 5% CO_2_ in the following high-glucose Dulbecco’s modified Eagle’s medium (Wako, Osaka, Japan) containing 10% fetal bovine serum (BioWest, Logan, UT, USA), 200 units/mL penicillin, 10 mg/mL streptomycin, and 25 mg/mL amphotericin B (Cosmo Bio Company, Tokyo, Japan). Lenvatinib was purchased from Cayman Chemical Company (Ann Arbor, MI, USA).

### 2.4. Animals

We established a mouse model of HCC according to a previous description by Matsuki et al. [20]. Four-week-old female athymic mice (BALB/c nu/nu) were purchased from Charles River Laboratories Japan, Inc. (Kanagawa, Japan) and used at 5 weeks of age. The mice were housed at 22 °C under a 12-h light/dark cycle with free access to food and water. All experimental procedures using animals were performed according to the guidelines of the Animal Care Committee at Kansai Medical University, and this study was approved by the Institutional Review Board of our hospital (approval no. 21-058). The number of animals used was kept to a minimum, and maximum effort was made to prevent suffering. The tumor dimensions were measured three times (on days 0, 7, and 14) using calipers, and the tumor volume was calculated as follows: ½ × length × width^2^.

### 2.5. Treatment of Subcutaneous Tumor Models

HuH-7 cells (5 × 10^6^/100 μL) were subcutaneously inoculated into the right flank of the mice. When the mean tumor volume reached approximately 180 mm^3^, the mice were randomly allocated to treatment groups (day 0). On days 7 and 14, the tumor volume and body weight were measured and blood samples were obtained.

The sheet insertion procedure was as follows: each mouse was subjected to deep anesthesia by intraperitoneal injection of a mixture of 4 mg/kg midazolam, 5 mg/kg butorphanol, and 0.75 mg/kg medetomidine. The skin in the immediate vicinity of the tumor was incised, the subcutaneous area was thoroughly dissected, and one or two sheets (10 × 10 mm) were inserted. The wound was closed and repaired using three stitches of 6-0 proline nodal suture.

To compare orally administered lenvatinib with lenvatinib sheets, the mice were categorized into four groups according to the treatment: the no treatment group (no treatment: mice were only followed), the lenvatinib oral group (oral [3 mg]: mice received 10 mg/kg/day lenvatinib, orally administered daily for a total of 3.01 mg/14 days), the lenvatinib 1 mg release sheet group (mice received 1 mg sheets, which is 32.5% of the oral dose of 10 mg/kg/day lenvatinib), and the lenvatinib 2 mg release sheet group (mice received 2 mg sheets, which is 65% of the oral dose of lenvatinib). Two 1 mg lenvatinib sheets inserted in layers were defined as the 2 mg sheet group. Lenvatinib was dissolved in 3 mmol/L HCl and orally administered to each mouse once daily for the indicated doses for 14 days.

To compare the sheet positions, the mice were categorized into four groups according to the treatment: the control operation group (control), in which drug-free sheets were inserted directly above the subcutaneous tumor, the direct group (1 mg sheets were inserted directly above), the peripheral group (the sheets were inserted peripherally), and the contralateral group (the sheets were inserted contralateral to the subcutaneous tumor).

### 2.6. Peritoneal Metastasis Tumor Therapy in Mice

The mice were randomly divided into two groups according to the treatment: drug-free sheet (control) and 1 mg lenvatinib sheet (1 mg sheet) groups. After the intraperitoneal injection of HuH-7-Luc cells (5 × 10^7^/500 µL), the sheets were subcutaneously inserted into the right flank of the mice. The appropriate number of cells administered was adjusted using HuH-7-Luc cells to establish a peritoneal seeding model. Luminescence was analyzed to confirm if peritoneal seeding had occurred. The day when the sheet was inserted was set as day 0, and the overall survival period was recorded.

### 2.7. Histological Examination

Xenograft tumors were collected on the day after the last administration. The tumor fragments were fixed in 10% formalin and embedded in paraffin. Sections (approximately 4-µm thick) were prepared from each block and subjected to hematoxylin and eosin (H&E) staining and immunostaining for CD31. Staining of endothelial cells and measurements of microvessel density (MVD) using anti-CD31 antibodies were performed using a modified version of a previously described method [21]. Briefly, anti-mouse endothelial cell marker CD31 rat monoclonal antibody (1:300) (Dianova GmbH, Hamburg, Germany) was used as the primary antibody, and horseradish peroxidase-conjugated HISTOFINE simplestain MAX-PO (Rat) (Nichirei Bioscience, Tokyo, Japan) was used as the secondary antibody for staining the tissue sections according to the above-mentioned method. The positive sites were visualized as brown color using 3,3’-diaminobenzidine. Five fields of view, excluding necrotic areas, were selected and measured using WinROOF (version 7.2) (Mitani Corporation, Fukui, Japan). The mitotic index (MI) was measured as the percentage of mitotic nuclei with loss of nuclear membrane and chromosome formation among 1000 tumor cells (excluding stromal cells, blood vessels, and leukocytes) in H&E-stained specimens observed at 400× magnification. The vascular area ratio was calculated by measuring CD31-positive areas (μm^2^) in the field of view of 200× magnification in five views and dividing the average by the area of one field of view (376,496.9 μm^2^). MVD was expressed as the number of microvessels per 1 mm^2^ and calculated as the average of five fields of view. The internal diameter of vessels was calculated by randomly selecting 30 CD31-positive capillaries with clearly observable lumens and measuring the internal diameter of the lumen without the vascular endothelium at 400× magnification.

### 2.8. Quantification of the Serum Lenvatinib Levels

The serum lenvatinib levels were measured according to a previously described method [22] with some modifications. Serum samples were obtained from the abdominal aorta of the mice. To extract lenvatinib from the serum samples, 50 µL of serum was diluted with 150 µL of ultrapure water, and 800 µL of cold acetone (Wako) was added. Then, the samples were placed at −20 °C for 1 h and centrifuged at 15,000 rpm for 10 min. The supernatant was dried using a CC-105 centrifugal concentrator (TOMY, Tokyo, Japan) and TU-1000 low temperature trap (TOMY). The dried sample was dissolved in 50 µL of 20% acetonitrile in water and centrifuged at 15,000 rpm for 1 min, and the supernatant was analyzed using liquid chromatography and mass spectrometry (LC-MS/MS). LC was performed using a Prominence HPLC system (Shimadzu Corporation, Kyoto, Japan), and the chromatographic system was operated using Analyst (version 1.7.1; SCIEX, Framingham, MA, USA). Chromatographic separations were performed using a C18 column (SCIEX, dimensions 5 μm, 4.6 mm × 150 mm) at 40 °C. The mobile phase was composed of two solvents (solvent A, 0.1% trifluoroacetic acid [Wako] in ultrapure water; solvent B, 0.1% trifluoroacetic acid in acetonitrile (Merck, Darmstadt, Germany). Then, 10 µL of the samples was injected into the column, separated at a flow rate of 0.5 mL/min, and eluted in a linear gradient. MS was performed using the API3200 LC-MS/MS System (SCIEX). The retention time of lenvatinib was approximately 8.7 min under the aforementioned elution conditions. The serum lenvatinib level was quantified as the count of the peak area at the intended retention time. Because the recovery rate of drug in acetone solution is equal for each serum sample, the amount of lenvatinib detected was used as a relative amount to compare the drug amounts between the samples.

### 2.9. Statistical Analysis

Data were presented as means ± standard deviation (SD). The tumor volume and body weight were compared between the treatment groups using the Steel–Dwass multiple comparison test. The MI, vascular area ratio, MVD, and internal diameter of vessels were compared between the treatment groups using one-way analysis of variance, followed by multiple comparisons using the Tukey test. Overall survival was estimated using the Kaplan–Meier method and compared using the log-rank test. Differences with *p*-values of <0.05 were considered statistically significant. Statistical analysis was performed using R (version 3.4.3; R Foundation for Statistical Computing, Vienna, Austria) and Prism9 (GraphPad Software Inc., La Jolla, CA, USA).

## 3. Results

### 3.1. PCL Nanofiber Sheets Release Lenvatinib in a Sustainable Manner

The diameter of the PCL-only fibers (1555 ± 538 nm; *n* = 50) exhibited large dispersion. In contrast, the diameter of nanofibers containing lenvatinib (683 ± 160 nm; *n* = 50) had high uniformity and produced thinner fibers than the PCL-only fibers (Figure 2A). Both fibers, with and without lenvatinib, formed smooth, uniform fibers without beads. Also, both crystallization temperature and melting temperature were not affected by the incorporation of lenvatinib, suggesting that lenvatinib uniformly disperses in the amorphous region of PCL (Appendix A). These observations are very important because the morphology and crystallinity significantly influence the sustained release of lenvatinib. The PCL nanofiber sheets were fabricated by electrospinning under a previously optimized condition [15]. The concentration of PCL was carefully selected because its viscosity has a dominant effect on the jet behavior. The nanofiber sheet continuously released lenvatinib for more than 8 weeks without an initial burst release, and 14.85 ± 0.86%, 19.15 ± 0.73%, and 28.02 ± 2.15% of lenvatinib were released after 2, 4, and 8 weeks, respectively (Figure 2B and Table 1). The results showed that approximately 1 mg was released from the sheets prepared in this study in 2 weeks, which is equivalent to approximately 32.5% of the 10-mg/kg/day (3.01 mg/14 days) oral dose of lenvatinib in mice (average weight of 21.5 g).

### 3.2. Nanofibrous Sheets Incorporating Lenvatinib Exhibited Antitumor Effects

We compared the antitumor effects of orally administered lenvatinib with those of lenvatinib sheets in a subcutaneous tumor model. On day 7 after the start of treatment, both the 1 mg (tumor volume: 354 ± 55 mm^3^) and 2 mg (344 ± 38 mm^3^) sheet groups showed significant tumor growth inhibition compared with the no treatment (1102 ± 138 mm^3^; *p* < 0.01) and oral (787 ± 128 mm^3^; *p* < 0.05) groups. On day 14, the oral (780 ± 111 mm^3^) group showed statistically significant inhibition compared with the no treatment (2049 ± 313 mm^3^; *p* < 0.01) group, and the 1 mg (375 ± 54 mm^3^) and 2 mg (315 ± 67 mm^3^) sheet groups showed significant inhibition compared with the no treatment (*p* < 0.01) and oral (*p* < 0.05) groups (Figure 3A and Table 2). Weight loss was not observed in any group (Figure 3B). The aspartate transaminase (IU/L ± SD) and lactate dehydrogenase (IU/L ± SD) levels in each group on day 14 were as follows: no treatment group, 182 ± 89 IU/L and 1417 ± 864 IU/L; oral group, 115 ± 20 IU/L and 608 ± 117 IU/L; and 1 mg sheet group: 90 ± 26 IU/L and 441 ± 197 IU/L, respectively (Table 2). Because the antitumor effect was not different between the 1 mg and 2 mg sheet groups, we planned the subsequent experiments using the lower volume 1 mg sheets.

### 3.3. Histopathological Analysis

The subcutaneous tumors in the no treatment, oral, and 1 mg sheet groups were examined histologically 14 days after the start of treatment. Tumor sections stained with H&E in each group (Figure 4A–C; 400×), representative images of vessel walls stained with CD31 (Figure 4D–F; 200×), and cross-sections of the vessels (4G–I; 400×) are shown. Tumor angiogenesis was assessed by measuring the area of CD31-positive areas per unit area (vessel area ratio), MVD (number of microvessels/mm^2^), and diameter of blood vessels (μm) after staining with anti-CD31 antibody. The proliferative potential of the cells was assessed using MI (%) (Table 3). The proliferative potential was inhibited in the oral (2.88 ± 0.36%; *p* < 0.01) and 1 mg sheet (1.96 ± 0.36%; *p* < 0.01) groups compared with that in the no treatment (4.14 ± 0.63%) group. Furthermore, the proliferative capacity of the 1 mg sheet group was more inhibited than that of the 3 mg oral group (Figure 4J). The vascular area ratio was reduced in the oral (1.92 ± 0.26%; *p* < 0.01) and 1 mg sheet (0.97 ± 0.43%; *p* < 0.01) groups compared with that in the no treatment (4.10 ± 1.34%) group (Figure 4K). Furthermore, MVD was significantly reduced in the 1 mg sheet (243 ± 52/mm^2^; *p* < 0.05) group compared with that in the no treatment (362 ± 48/mm^2^) group, despite the lower drug amount than that in the oral (336 ± 68/mm^2^) group (Figure 4L). The mean diameter of the vessels was also significantly smaller in the oral (3.53 ± 0.47 μm; *p* < 0.01) and 1 mg sheet (3.52 ± 0.60 μm; *p* < 0.01) groups than that in the no treatment (7.29 ± 1.64 μm) group (Figure 4M).

### 3.4. Lenvatinib Sheets Maintain the Serum Drug Level

Relative serum lenvatinib levels were determined using LC techniques and compared between the oral and 1 mg sheet groups. The serum lenvatinib levels in the oral group peaked at 3 h after administration and disappeared at 24 h, whereas the serum level peaked at 24 h and then maintained a gradual downward trend until day 14 in the 1 mg sheet group (Figure 5). Because lenvatinib was administered daily in the oral group, a peak level should be observed every day. However, it should be noted that the blood samples collected after day 3 were taken 24 h after the last oral dose; thus, the apparent blood level was zero.

### 3.5. The Antitumor Effect of Lenvatinib Sheets in Different Insertion Positions

The mice were categorized into four groups according to the treatment: the control, direct, peripheral, and contralateral groups (Figure 6A). On day 7 after the start of treatment, the direct (tumor volume ± SD: 247 ± 61 mm^3^; *p* < 0.01), peripheral (340 ± 67 mm^3^; *p* < 0.01), and contralateral (256 ± 62 mm^3^; *p* < 0.01) groups showed significant inhibition of tumor volume compared with the no treatment (1233 ± 333 mm^3^) group. Similar results were observed on day 14, with significant inhibition of tumor growth in the direct (377 ± 41 mm^3^; *p* < 0.01), peripheral (431 ± 67 mm^3^; *p* < 0.01), and contralateral (397 ± 48 mm^3^; *p* < 0.01) groups compared with that in the no treatment (2003 ± 327 mm^3^) group (Figure 6B and Table 4). No significant differences in tumor volume were observed between the three sheet positions on days 7 and 14. Although we compared the serum levels of lenvatinib in the three groups with different sheet insertion positions, no significant difference on days 7 and 14 was observed between the three groups (Figure 6C). Moreover, there were no AEs of weight loss in each group (Figure 6D).

### 3.6. Lenvatinib Sheets Improved Survival in a Mouse Peritoneal Seeding Model

The overall survival was significantly prolonged in the 1 mg sheet group compared with that in the control group (*p* < 0.01) (Figure 7). In the control group, gross peritoneal tumor nodules became visible on day 14 after treatment, and six of eight mice (75%) exhibited bloody ascites on day 20. On day 30 after treatment, seven of eight mice (87.5%) died. In contrast, all mice in the 1 mg sheet group remained alive 30 days after treatment, and none of them exhibited peritoneal disseminated nodules or ascites.

## 4. Discussion

PCL sheets are prepared by electrospinning of PCL to resemble an unwoven fabric of ultrafine fibers and are thin, flexible, and pliable nanofiber sheets with a thickness of approximately 0.1 mm [16]. They can be easily cut into the required size for insertion or application, making them an ideal material for clinical implementation. Because the drug is uniformly contained in each ultrafine fiber, it can be released at a constant rate over a long duration (Figure 1 and Figure 2). When inserted into the body, PCL is designed to biodegrade slowly over a period of more than 1 year due to its own hydrolysis, thereby eliminating the need for removing the sheet after surgery. Recently, a PCL sheet containing vitamin B12 was developed, and when this sheet was implanted around the injured area in a rat model of sciatic nerve injury, nerve axons regenerated and motor and sensory functions were restored within 6 weeks after surgery [23]. Furthermore, in a rat model of sciatic nerve transection, nerve conduction velocity was restored 8 weeks after surgery [18]. Currently, a clinical trial is underway to insert vitamin B12-containing PCL sheets into patients requiring carpal tunnel release and nerve sutures. Remarkably, we could only find a few studies on PCL sheets containing molecular-targeted drugs. We developed lenvatinib-incorporating sheets that are effective in an HCC mouse model. Initially, we expected that the PCL sheets would have a local sustained-release effect and conducted experiments assuming that they would be applied to the cancer site. In clinical applications, they were conceived to be applied to the surface of liver tumors when curative resection was judged to be difficult or to be applied directly to the tumor thrombus in the portal or hepatic vein. However, in these clinical applications, concerns regarding the misalignment of the attached sheet and damage to blood vessels arose. However, in this study, the serum levels of lenvatinib were maintained, regardless of the position of the inserted sheet, and the results showed an adequate antitumor effect. Therefore, in future clinical applications, we suggest that the sheet will show an antitumor effect when applied to the body surface.

The dosage of lenvatinib for human HCC is eight or twelve mg/day, which is equivalent to 10–15 mg/kg/day for mice [20]. In this study, we prepared sheets containing 6.6 mg lenvatinib with a sustained release of approximately 1 mg in 14 days. Moreover, 1 mg of lenvatinib is equivalent to 32.5% of the 3.01 mg dose of lenvatinib (10 mg/kg/day) orally administered to mice with a body weight of 21.5 g for 14 days. We found that sheets containing 1 mg lenvatinib, which is 32.5% of the oral dosage, showed sufficient antitumor effects (Figure 3 and Figure 4). By incorporating the drug in the sheets, the serum lenvatinib levels were maintained over a period of time, and the antitumor effect was likely more significant than that of oral administration (Figure 3, Figure 4 and Figure 5). As shown in Figure 5, the serum drug level in the 3 mg oral group disappeared after 1 day. However, the reason why the 3 mg oral group showed an antitumor effect in Figure 3 and Figure 4, is because the blood sample was collected after day 3, 24 h after the last oral dose of lenvatinib. Because lenvatinib is administered orally daily, it is likely that its blood levels increased early after oral administration, indicating an antitumor effect. Furthermore, lenvatinib sheets maintained stable blood levels in a subcutaneous tumor mouse model, regardless of the position of the sheets (Figure 6). The lenvatinib sheets also had a prolonged survival effect 30 days after the start of treatment in a peritoneal seeding model (Figure 7). An antitumor effect was exhibited because lenvatinib was efficiently absorbed from the capillaries in the dermis by exhibiting a stable sustained-release effect of the drug, which is a characteristic of the sheets. The proposed mechanism of action on the disseminated nodules was that lenvatinib in the circulating blood exerted its effects through the capillaries of the peritoneum. Furthermore, because mice have a thin subcutaneous tissue layer and the dermis and peritoneum are located close to each other, it is possible that the slow release of lenvatinib from the sheet acted directly on the disseminated nodule through the peritoneum. 

Paclitaxel is used for intraperitoneal chemotherapy, which is being clinically studied as a peritoneal dissemination treatment for gastric cancer. Because paclitaxel is lipophilic and has a large molecular weight (853.906 g/mol), it is not absorbed through peritoneal capillaries and maintains a high concentration in ascites. The antitumor effect of paclitaxel has been suggested to be exerted through its penetration into the disseminated lesion [24]. 

In contrast, lenvatinib is a water-soluble drug with approximately half the molecular weight (426.853 g/mol) as that of paclitaxel, and theoretically, it is easily absorbed in the capillaries of the peritoneum. Therefore, by placing the sheet directly above the peritoneum, lenvatinib may have acted directly on the seeding model through the peritoneum or by its transfer to the ascites. Although lenvatinib is a concentration-dependent drug, its plasma trough concentration and the area under the blood concentration–time curve are associated with the occurrence of AEs, leading to early dose reduction or early discontinuation [8,25]. In a previous study, the incidence of liver dysfunction and thrombocytopenia increased when the plasma trough concentration of lenvatinib exceeded 88 ng/mL in patients with thyroid cancer [26]. 

In this study, the reduced dose of lenvatinib and the sustained-release properties of the sheets were used to help maintain the serum level of lenvatinib and exert an antitumor effect without causing side effects by suppressing the rapid increase in blood levels of the drug (Figure 5). Although avoidance of AEs could not be directly evaluated in these experiments, it is possible that decreasing the total drug dose led to a reduction in the incidence of AEs, and future experiments are warranted. The clinical application of PCL sheets will likely increase with the development of a method for adjusting the release of drugs from the sheet from outside of the body or one for easily removing the sheets by devising their shape. Presently, remote-control methods for DDS using ultrasonic waves [27], magnetic fields [17], thermal neutrons [15], and near-infrared rays [28], among others, have been reported. In cancer chemotherapy, Abraxane^®^ (albumin-added paclitaxel) [29], liposomal formulations containing irinotecan [30] and daunorubicin/cytarabine, and ADCs [11] have been commercialized, and other micelle formulations [10] are under development. Because PCL sheets have a sustained-release property and can avoid a rapid increase in the blood concentration of the drug after administration, a reduction of drug-induced side effects is possible. Our results may enable the development of sheets containing multiple anticancer agents, such as cytotoxic chemotherapeutic agents, molecular-targeted agents, and microRNAs. Furthermore, we can consider the development of new therapeutic devices, such as skin patches, as well as the subcutaneous insertion of sheets. Our study highlighted that drug-encapsulating PCL sheets could serve as a new and innovative DDS.

## 5. Conclusions

The lenvatinib-incorporating PCL sheets maintained the serum levels of lenvatinib and showed adequate antitumor effects in an HCC mouse model. This study showed that combining PCL with lenvatinib is possible and that lenvatinib-incorporating PCL sheets could serve as a novel DDS.

## Figures and Tables

**Figure 1 nanomaterials-12-01364-f001:**
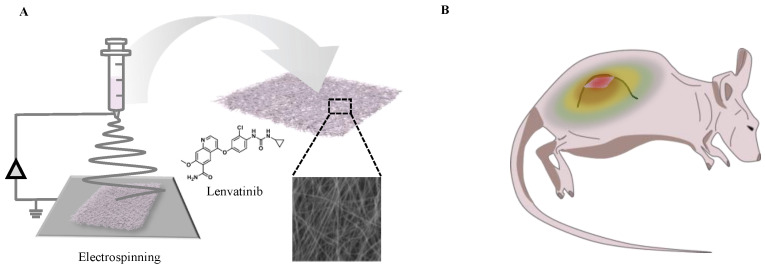
(**A**) Design concept for a smart nanofiber system that incorporated lenvatinib using the electrospinning method. (**B**) The nanofiber allows for the sustained release of lenvatinib at the local tumor site through diffusion. Localized administration of molecular-targeted drugs can improve the therapeutic efficiency.

**Figure 2 nanomaterials-12-01364-f002:**
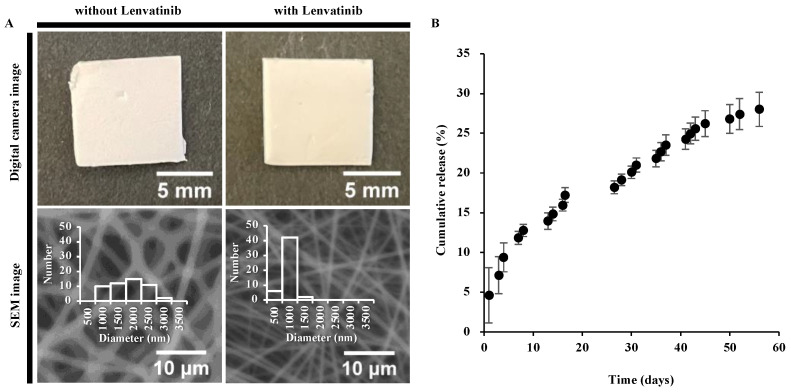
(**A**) Digital camera image and scanning electron microscopy image of lenvatinib unloaded and loaded nanofibers. The fiber diameters were 1555 ± 538 nm and 683 ± 160 nm, respectively. (**B**) Lenvatinib release profile from the nanofiber sheet.

**Figure 3 nanomaterials-12-01364-f003:**
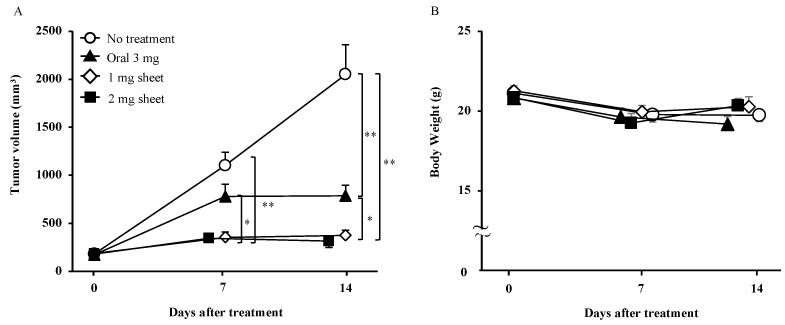
Antitumor effect of lenvatinib sheets in a subcutaneous tumor model. In total, 5 × 10^6^ HuH-7 cells were subcutaneously transplanted into BALB/c nude mice. The mice were categorized into four groups according to the treatment: the no treatment (open circle; day 0: *n* = 28, day 7: *n* = 16, and day 14: *n* = 18), oral (filled triangles; day 0: *n* = 20, day 7: *n* = 10, and day 14: *n* = 10), 1 mg sheet (open diamond shape; day 0: *n* = 30, day 7: *n* = 15, and day 14: *n* = 20), and 2 mg sheet (filled square; day 0: *n* = 20, day 7: *n* = 10, and day 14: *n* = 10) groups. (**A**) Time course of xenograft tumor volume. (**B**) Body weight in each group. Data are presented as means ± standard deviations. The *p*-values between the indicated groups are presented as * *p* < 0.05 and ** *p* < 0.01. The results are the sum of three independent experiments.

**Figure 4 nanomaterials-12-01364-f004:**
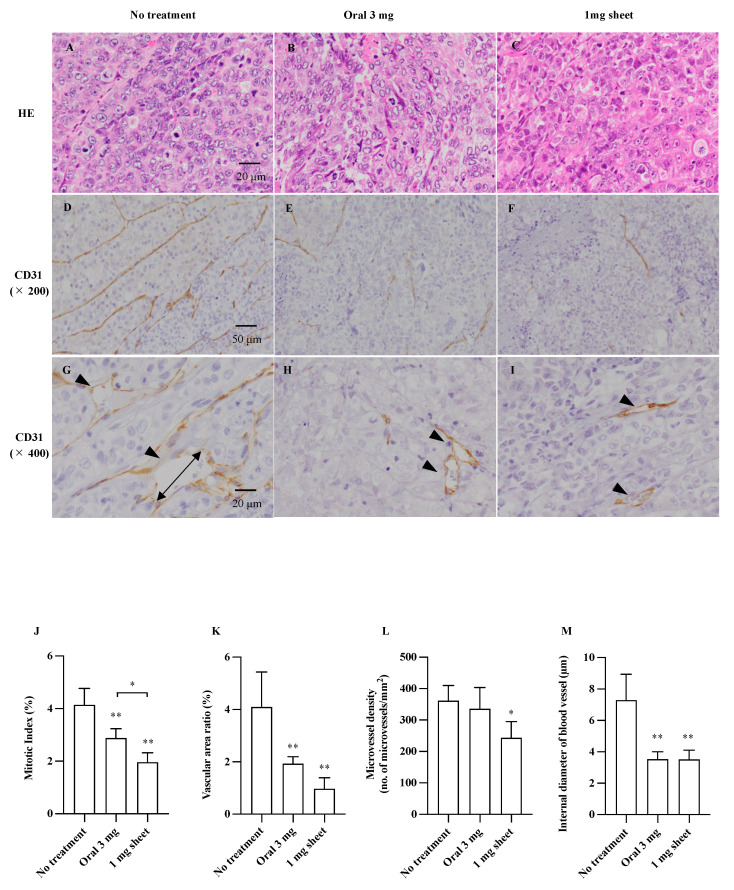
Immunohistochemical analysis of subcutaneous tumors. Formalin-fixed paraffin-embedded tumor sections were stained with H&E (**A**–**C**; 400×, bar = 20 μm) and anti-CD31 antibody (**D**–**I**). Representative images of vessel walls stained with CD31 (**D**–**F**; 200×, bar = 50 μm), and cross-section of vessel (**G**–**I**; 400×, bar = 20 μm). Arrowheads indicate microvessels. Quantification of MI (**J**), vascular area ratio (**K**), MVD (**L**), and internal diameter of vessels (**M**). Data are presented as means ± standard deviations (*n* = 5). The *p*-values between the indicated groups are presented as * *p* < 0.05 and ** *p* < 0.01.

**Figure 5 nanomaterials-12-01364-f005:**
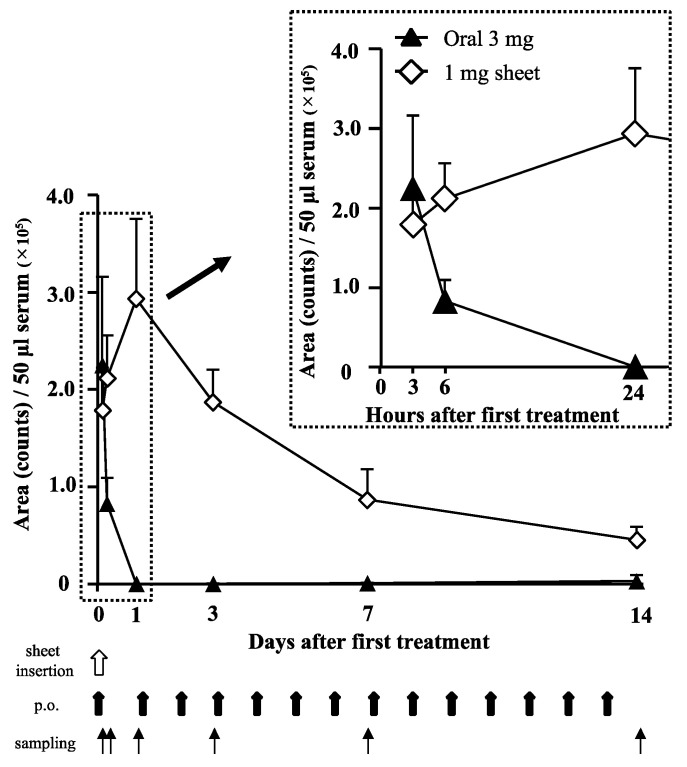
Trends of serum lenvatinib levels. The serum lenvatinib levels in the oral (filled triangle; 3 h: *n* = 5, 6 h: *n* = 5, day 1: *n* = 5, day 3: *n* = 5, day 7: *n* = 10, and day 14: *n* = 10) and 1 mg sheet (open diamond shape; 3 h: *n* = 5, 6 h: *n* = 5, day 1: *n* = 5, day 3: *n* = 5, day 7: *n* = 10, and day 14: *n* = 10) groups were analyzed using LC-MS/MS (at least five mice in each group were tested at each time). Blood sampling was attempted at 3, 6, and 24 h and 3, 7, and 14 days after sheet insertion or the first oral administration. The report of the animal experiment in the developed pharmaceutical company is described below (https://www.pmda.go.jp/drugs/2015/P201500005/170033000_22700AMX00640_I100_1.pdf, accessed January 6, 2022). The time to reach the maximum level of lenvatinib in mice was 0.5–1 h, and the elimination half-life in the blood was 1.74–2.09 h; the original peak was much higher.

**Figure 6 nanomaterials-12-01364-f006:**
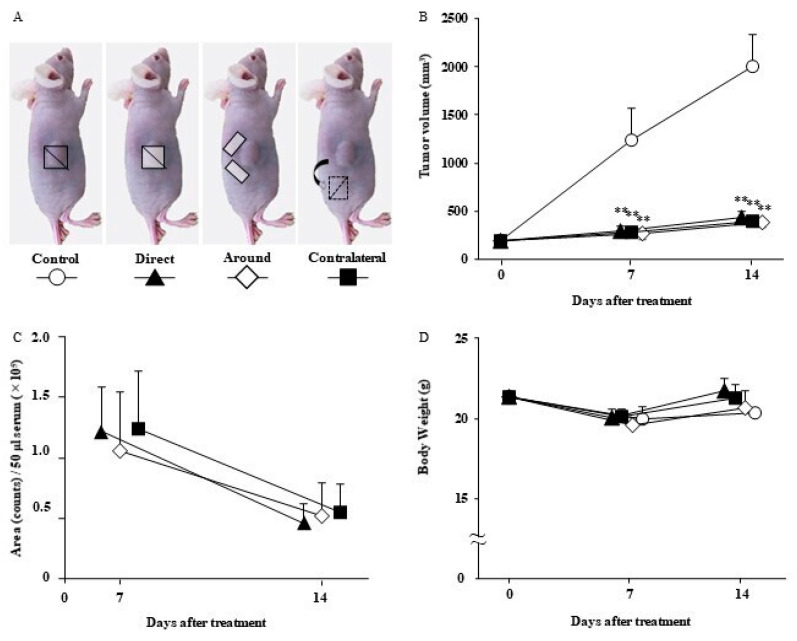
Antitumor effect of lenvatinib sheets in the different insertion positions. The mice were categorized into four groups according to the treatment method: the control (open circle; day 0: *n* = 11, day 7: *n* = 6, and day 14: *n* = 11), direct (filled triangle; day 0: *n* = 10, day 7: *n* = 5, and day 14: *n* = 10), peripheral (open diamond shape; day 0: *n* = 10, day 7: *n* = 5, and day 14: *n* = 10), and contralateral (filled square; day 0: *n* = 10, day 7: *n* = 5, and day 14: *n* = 10) groups. (**A**) Schematic diagram of the mice. (**B**) Time course of xenograft tumor volume. (**C**) Trends of serum lenvatinib level. (**D**) Body weight in each group. Data are presented as means ± standard deviations. ** *p* < 0.01 compared with the control group. The results are the sum of two independent experiments.

**Figure 7 nanomaterials-12-01364-f007:**
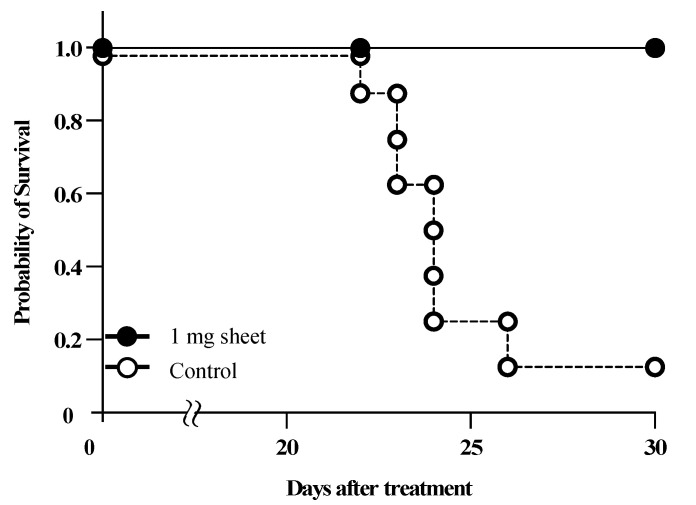
Effect of lenvatinib sheets on survival in a mouse peritoneal seeding model. In total, 5 × 10^7^ HuH-7-Luc cells were intraperitoneally transplanted into BALB/c nude mice, and simultaneously, the sheets were inserted subcutaneously. The Kaplan–Meier survival curves represent the cumulative survival of the following treatment groups in one experiment: control (*n* = 8; open circles) and 1 mg sheet (*n* = 10; filled circles) groups. *p* < 0.001 compared with the control group (by the log-rank test).

**Table 1 nanomaterials-12-01364-t001:** Cumulative release rate of lenvatinib.

2 Weeks	4 Weeks	8 Weeks
14.85 ± 0.86%	19.15 ± 0.73%	28.02 ± 2.15%

**Table 2 nanomaterials-12-01364-t002:** Tumor volume and markers of liver injury in the serum.

	Tumor Volume (mm^3^)		Aspartate Transaminase (IU/L)	Lactate Dehydrogenase (IU/L)
	Day 7	Day 14	Day 14	Day 14
No treatment	1102 ± 138	2049 ± 313	182 ± 89	1417 ± 864
Oral 3 mg	0787 ± 128	0780 ± 111	115 ± 20	0608 ± 117
1 mg sheet	354 ± 55	375 ± 54	090 ± 26	0441 ± 197
2 mg sheet	344 ± 38	315 ± 67	—	—

**Table 3 nanomaterials-12-01364-t003:** Immunohistochemical data analysis.

	Mitotic Index	Vascular Area Ratio	Microvessel Density (mm^2^)	Internal Diameter of Vessels (μm)
No treatment	4.14 ± 0.63%	4.10 ± 1.34%	362 ± 48	7.29 ± 1.64
Oral 3 mg	2.88 ± 0.36%	1.92 ± 0.26%	336 ± 68	3.53 ± 0.47
1 mg sheet	1.96 ± 0.36%	0.97 ± 0.43%	243 ± 52	3.52 ± 0.60

**Table 4 nanomaterials-12-01364-t004:** Tumor volume in different insertion positions.

	Tumor Volume (mm^3^)	
	Day 7	Day 14
Control	1233 ± 333	2003 ± 327
Direct	247 ± 61	377 ± 41
Peripheral	340 ± 67	431 ± 67
Contralateral	256 ± 62	397 ± 48

## Data Availability

Not applicable.

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
