# Peer review of "Efficacy of Nanofiber Sheets Incorporating Lenvatinib in a Hepatocellular Carcinoma Xenograft Model"

_nanomaterials, 2022, doi:10.3390/nano12081364_

Round 1

Reviewer 1 Report

The present article analyses the anticancer efficacy of poly(ε-caprolactone) (PCL) sheets encapsulating lenvatinib in a murine subcutaneous hepatocellular carcinoma model. The study compared the anticancer effects, the serum drug concentrations and different histopathological parameters of tumour specimens after treatments either with the chemotherapeutic drug lenvatinib administered alone or with lenvatinib encapsulated in PCL sheets.

Broad comments 

The article extensively studies the efficacy of treatments on a HCC tumor model with the orally administered lenvatinib alone at the concentration of 3 mg and lenvatinib incorporated in PCL sheets (1 and 2 mg) as new drug delivery system. Data obtained demonstrated that this new DDS was very effective in inhibiting tumour growth with respect to untreated or free lenvatinib-treated mice. When histopathological parameters were analysed, the mitotic index, the ratio of vascular area and the microvessels density resulted to be lower in lenvatinib-sheet treated mice when compared to free drug-treated or control mice.

In addition, lenvatinib sheets allowed to maintain drug serum concentrations for a longer time (14 days), independently of the site of the insertion (direct, peripheral or contralateral). This finding, along with the higher antitumor efficacy achieved with a lower drug concentration through PCL sheets encapsulating lenvatinib (as compared with orally administrated free drug), strongly suggest that PCL sheets could be a promising system for a sustained deliver of different kinds of anticancer agents.

Author Response

We thank Reviewer 1 for the valuable comments.

Reviewer 2 Report

This study aims to develop a PCL sheets which encapsulated cancer drug- lenvatinib as a drug delivery system for hepatocellular carcinoma treatment. The results revealed the ability of PCL sheets to achieve lenvatinib delivery on mouse subcutaneous tumor model. There are some comments to this manuscript:

  1. In materials and methods section 2.4, the authors mention that tumor dimensions were measured two or three times weekly. However, the tumor volume results only showed three observation. The interval of tumor volume observation was not appropriate, please clarify.
  2. I suggest that the authors indicate the day of drug administered for mouse.
  3. The serum lenvatinib concentration data in figure 5 indicated that the drug concentration of the group of oral 3 mg was disappeared after 1 day. However, the antitumor effect of Oral 3mg was still effective, the authors should discuss more in the discussion section.
  4. The scale bars in figure 4 are incorrect, the authors need to recheck the data.
  5. The authors should revealed how many animals were used in experiments.

Author Response

We thank Reviewer 2 for the valuable comments.

Major points

1. In materials and methods section 2.4, the authors mention that tumor dimensions were measured two or three times weekly. However, the tumor volume results only showed three observation. The interval of tumor volume observation was not appropriate, please clarify.

Response

Thank you very much for your helpful comments.

As you indicated, in the Materials and Methods section 2.4, lines 130–131 are incorrect, and tumor volumes were measured three times, on days 0, 7, and 14.

We have changed the following sentences in the Materials and Methods section.

Materials and Methods 2.4 (lines 130–131)

“The tumor dimensions were measured three times (on days 0, 7, and 14) using calipers, and the tumor volume was calculated as follows: ½ × length × width2.”

(please see the attachment)

2. I suggest that the authors indicate the day of drug administered for mouse.

Response

Thank you very much for your excellent suggestion.

In the Materials and Methods section 2.5, lines 145–146, we stated that they are administered daily. In addition, we have changed Figure 5 to show the date of oral administration, for clarification.

Materials and Methods 2.5 (lines 145–146)

“the lenvatinib oral group (oral [3 mg]: mice received 10 mg/kg/day lenvatinib, orally administered daily for a total of 3.01 mg/14 days)”

3. The serum lenvatinib concentration data in figure 5 indicated that the drug concentration of the group of oral 3 mg was disappeared after 1 day. However, the antitumor effect of Oral 3mg was still effective, the authors should discuss more in the discussion section.

Response

Thank you very much for your invaluable comments.

As noted in the Results section 3.4, lines 311–313, blood samples were obtained after day 3, 24 h after the last oral dose of lenvatinib, and thus, the blood concentration was zero. However, because lenvatinib was administered orally daily, we consider that its blood concentration increased early after oral administration, thereby showing an antitumor effect.

We have added the following sentences to the Discussion section (lines 399–405).

By incorporating the drug in the sheet, the serum lenvatinib levels were maintained over a period of time, and the antitumor effect was likely more significant than that of oral administration (Figs. 3, 4 and 5). As shown in Fig. 5, the serum drug level in the 3 mg oral group disappeared after 1 day. However, the reason why the 3 mg oral group showed an antitumor effect in Figs. 3 and 4, is because the blood sample was collected after day 3, 24 h after the last oral dose of lenvatinib. Because lenvatinib is administered orally daily, it is likely that its blood levels increased early after oral administration, indicating an antitumor effect.

4. The scale bars in figure 4 are incorrect, the authors need to recheck the data.

Response

Thank you very much for your important comments.

The scale in Figure 4A was incorrect and has been changed (from 50 to 20 μm).

5. The authors should revealed how many animals were used in experiments.

Response

Thank you for your recommendation.

The number of animals in each group has been added to the legend in Figures 3, 5, and 6 (lines 264–267, 315–318, and 342–345, respectively). We have removed the same information from the Materials and Methods section 2.5.

Reviewer 3 Report

This manuscript reports the loading of lenvatinib in polycaprolactone (PCL) fibers organized as sheets for the controlled release of the drug for the treatment of hepatocellular carcinoma. Sheet containing the drug were prepared and characterized. Around 30% of the drug is released in 56 days. In vivo, the presence of the drug-loaded device was more efficient than an oral dosing. It is also demonstrated that the drug is resorbed in plasma to reach the target site. The manuscript contains some imprecisions and should be revised for publication consideration.

The introduction should clearly mention if a systemic or local effect is desired.

From the results, it seems that a systemic effect is obtained (the drug is found in the serum). How can a device releasing the drug in the bloodstream avoid the adverse effects as stated in the introduction? None of the adverse effects tested were different from the controls.

Was the whole amount of lenvatinib associated with the fibers?

Were crystals of the drug visible on SEM? A good quality picture of unloaded and drug loaded film should be presented.

DSC data should be presented to have an idea of the state of the drug and the polymer in the final device.

What is the solubility of the drug in the releasing medium? Were the sink conditions respected?

LogP is 3.3 and solubility in water is 0.16 mg/ml (Cayman product information), sparingly soluble in water. May this explain that only 30% is released in 56 days? When is 100% of the drug released?

Why use HuH-7-Luc cells as there is no results of bioluminescence?

The numbers (data) given in the text of the results should be presented as tables for the ease of reading.

Not all the figures present the number of repetitions.

The quantification method for blood samples is not accurate. It should be presented as actual concentration of the drug in each sample with a reference to a standard curve. Line 291 presents “relative serum concentrations” relative to what? How was this data obtained?

The insert of figure 5 seems to present different data points than the larger picture. For example, 1 mg sheet at 24h, is above 3 in the large picture whereas it is under 3 in the insert. The SD look different as well.

In the discussion, line 350, it is mentioned that the drug is uniformly contained, but this was not demonstrated in the study.

English could be checked by an English-speaking scientist.

Author Response

Thank you for your valuable comments.

Major points

  1. The introduction should clearly mention if a systemic or local effect is desired.

Response

Thank you for your helpful recommendations.

The purpose of our study was to determine whether PCL nanofiber sheets incorporating lenvatinib could be a new DDS with systemic effects. To clarify our objective, we have added the word “systemic” to the Introduction section (lines 87–88).

  1. From the results, it seems that a systemic effect is obtained (the drug is found in the serum). How can a device releasing the drug in the bloodstream avoid the adverse effects as stated in the introduction? None of the adverse effects tested were different from the controls.

Response

Thank you very much for your invaluable comments.

As you pointed out, we could not directly compare the reduction in adverse effects between the lenvatinib sheet and the oral group. However, our experiments suggested that it is possible to reduce the amount of drug used while maintaining the antitumor effect by incorporating the drug into the sheet. Decreasing the total drug dosage may lead to a reduction in adverse effects, and further experiments are needed.

We have added the following sentences in the Discussion section (lines 435–437).

Although avoidance of AEs could not be directly evaluated in these experiments, it is possible that decreasing the total drug dosage led to a reduction in the incidence of AEs, and future experiments are warranted.

  1. Was the whole amount of lenvatinib associated with the fibers?

Response

Yes, all lenvatinib was electrospun into the nanofiber sheets.

  1. Were crystals of the drug visible on SEM? A good quality picture of unloaded and drug loaded film should be presented.

Response

The drug molecules were completely dissolved and dispersed in the fibers without crystals. Therefore, we could not observe the crystal structure using SEM/EDX.

  1. DSC data should be presented to have an idea of the state of the drug and the polymer in the final device.

Response

We have included the DSC data in Figure S1 (Supplementary Figure1; lines 229 and 468–469). Usually, PCL has crystallization and melting temperatures of approximately 30°C and 60°C, respectively. However, the endothermic peak for the drug could be hardly observed using the DSC equipment.

(Please see the attachment)

  1. What is the solubility of the drug in the releasing medium? Were the sink conditions respected?

Response                

Yes, the sink condition was respected. The solubility of lenvatinib is not so high. Therefore, it was slowly released from the fibers.

  1. LogP is 3.3 and solubility in water is 0.16 mg/ml (Cayman product information), sparingly soluble in water. May this explain that only 30% is released in 56 days? When is 100% of the drug released?

Response

As the reviewer mentioned, we could achieve sustained release due to the low solubility of the drug in water. We expect 100% of the drug to be released in 3–4 months, according to our knowledge.

  1. Why use HuH-7-Luc cells as there is no results of bioluminescence?

Response

We apologize for the insufficient explanation regarding the use of HuH-7-Luc cells. Preliminary experiments were performed to establish a peritoneal seeding model using HuH-7-Luc cells with an appropriate number of cells administered. Luminescence was checked to confirm that peritoneal seeding was occurring. The experiment shown in Figure 7 was performed with the number of cells determined based on the preliminary experiment. Because luminescence was not confirmed in this experiment, the results were not shown.

We have added the following sentences in the Materials and Methods section 2.6 (lines 162–164).

“The appropriate number of cells administered was adjusted using HuH-7-Luc cells to establish a peritoneal seeding model. Luminescence was analyzed to confirm if peritoneal seeding had occurred.”

  1. The numbers (data) given in the text of the results should be presented as tables for the ease of reading.

Response

Thank you very much for your helpful recommendations.

The figures (data) listed in the text of the results were presented as Tables 1–4.

  1. Not all the figures present the number of repetitions.   

Response

Thank you very much for your invaluable comments.

Figure 3 is the sum of three experiments, Figure 6 is the sum of two experiments, and Figure 7 is the result of one experiment. We have added this information to the figure legends (lines 269–270, 348, 363).

  1. The quantification method for blood samples is not accurate. It should be presented as actual concentration of the drug in each sample with a reference to a standard curve. Line 291 presents “relative serum concentrations” relative to what? How was this data obtained?

Response

Thank you very much for your important comments.

Reviewer 3 requested us to present the actual serum concentration of lenvatinib in the animals. To determine the amount of drug in the serum, we treated the serum with cold acetone to minimize the contamination of serum proteins in the samples, recovered lenvatinib in the supernatant fraction, and analyzed it using LC-MS/MS. We assumed that the recovery of lenvatinib from serum samples was not 100% but expected that recovery rate of the drug in acetone solution would be equal for each serum sample. Unfortunately, we could not determine the recovery rate of lenvatinib using this method, and we indicated the amount of detected lenvatinib as a relative amount in the manuscript. Although this value was not the actual concentration, it allows the comparison of the amount of drug between samples.

We have added the following sentences in the Materials and Methods section 2.8 (lines 209–211).

Because the recovery rate of drug in acetone solution would be equal for each serum sample, the amount of lenvatinib detected was used as a relative amount to compare the drug amounts between the samples.

  1. The insert of figure 5 seems to present different data points than the larger picture. For example, 1 mg sheet at 24h, is above 3 in the large picture whereas it is under 3 in the insert. The SD look different as well.

Response

Thank you very much for your invaluable comments.

The insert and the large picture in Fig. 5 show the same data. However, the orientation and scale of the SD are different and may be misleading, as you have pointed out. Although the data is the same, we have edited Figure 5 to have an adjusted SD orientation, scale, and marker size.

  1. In the discussion, line 350, it is mentioned that the drug is uniformly contained, but this was not demonstrated in the study.

Response

We attempted to observe the drugs contained in the fiber using energy dispersive X-ray spectroscopy. However, we could not observe the crystal structures. Therefore, we concluded that the drug was uniformly dispersed in the fiber.

  1. English could be checked by an English-speaking scientist.

Response

Our manuscript has been proofread in English by an editing service again.

Round 2

Reviewer 3 Report

The revised manuscript is now of better quality, the methods are clearer and the results better presented.

Regarding the DSC data, it should be completed with the data from the drug alone, the mixture of both drug and polymer and the data from the sheet loaded with the drug, then it will give information about the structure of the drug in the device. Furthermore, information regarding the method should be given as well.

Author Response

We want to thank Reviewer #3 for providing valuable comments that helped improve the quality of the manuscript.

Comments from reviewer #3:

The revised manuscript is now of better quality, the methods are clearer and the results better presented.Regarding the DSC data, it should be completed with the data from the drug alone, the mixture of both drug and polymer and the data from the sheet loaded with the drug, then it will give information about the structure of the drug in the device. Furthermore, information regarding the method should be given as well.

Response :

We have added the DSC data of lenvatinib incorporated nanofiber, PCL nanofiber and free lenvatinib in Figure S@ (Supplementary Figure). Although lenvatinib only does not show significant endothermic peak in the range of the experiment, incorporation of lenvatinib decreases the crystallization temperature. This result indicates that lenvatinib may affect the crystallization of PCL. As for the mixture of both drug and polymer, we could not mix them without any solvent. So, we have only compare the PCL nanofibers with and without lenvatinib.

Figure S@. DSC curves of lenvatinib incorporated nanofibers, PCL nanofibers and lenvatinib.

Method

The thermal property of the PCL nanofibers was measured by differential scanning calorimetry (DSC) (DSC600, HITACHI, Tokyo, Japan) at a 15oC min-1 programming rate by using alumina pans. 

This manuscript is a resubmission of an earlier submission. The following is a list of the peer review reports and author responses from that submission.